# Locus Coeruleus Dysfunction and Trigeminal Mesencephalic Nucleus Degeneration: A Cue for Periodontal Infection Mediated Damage in Alzheimer’s Disease?

**DOI:** 10.3390/ijerph20021007

**Published:** 2023-01-05

**Authors:** Flavio Pisani, Valerio Pisani, Francesca Arcangeli, Alice Harding, Sim K. Singhrao

**Affiliations:** 1Programme Lead, MSc/MClinDent in Clinical Periodontology, Faculty of Clinical and Biomedical Sciences, School of Dentistry, University of Central Lancashire, Preston PR1 2HE, UK; 2I.R.C.C.S. “Santa Lucia” Foundation, Neurology and Neurorehabilitation Unit, Via Ardeatina, 306, 00179 Rome, Italy; 3Azienda Sanitaria Locale ASLRM1, Nuovo Regina Margherita Hospital, Geriatric Department-Advanced Centre for Dementia and Cognitive Disorders, Via Emilio Morosini, 30, 00153 Rome, Italy; 4Dementia and Neurodegenerative Disease Research Group, Faculty of Clinical and Biomedical Sciences, School of Dentistry, University of Central Lancashire, Preston PR1 2HE, UK

**Keywords:** locus coeruleus, mesencephalic trigeminal nucleus, trigeminal nerve, neurodegeneration, Alzheimer’s disease, periodontal disease, *Treponema denticola*

## Abstract

Alzheimer’s disease (AD) is a leading neurodegenerative disease with deteriorating cognition as its main clinical sign. In addition to the clinical history, it is characterized by the presence of two neuropathological hallmark lesions; amyloid-beta (Aβ) and neurofibrillary tangles (NFTs), identified in the brain at post-mortem in specific anatomical areas. Recently, it was discovered that NFTs occur initially in the subcortical nuclei, such as the locus coeruleus in the pons, and are said to spread from there to the cerebral cortices and the hippocampus. This contrasts with the prior acceptance of their neuropathology in the enthorinal cortex and the hippocampus. The Braak staging system places the accumulation of phosphorylated tau (p-tau) binding to NFTs in the locus coeruleus and other subcortical nuclei to precede stages I–IV. The locus coeruleus plays diverse psychological and physiological roles within the human body including rapid eye movement sleep disorder, schizophrenia, anxiety, and depression, regulation of sleep-wake cycles, attention, memory, mood, and behavior, which correlates with AD clinical behavior. In addition, the locus coeruleus regulates cardiovascular, respiratory, and gastrointestinal activities, which have only recently been associated with AD by modern day research enabling the wider understanding of AD development via comorbidities and microbial dysbiosis. The focus of this narrative review is to explore the modes of neurodegeneration taking place in the locus coeruleus during the natural aging process of the trigeminal nerve connections from the teeth and microbial dysbiosis, and to postulate a pathogenetic mechanism due to periodontal damage and/or infection focused on *Treponema denticola*.

## 1. Introduction

Alzheimer’s disease (AD) is a leading neurodegenerative disease represented by a clinical history in which changes in behavior and a failing memory are pronounced. Its final, conclusive diagnosis rests with the combination of clinical history and the presence of two neuropathological hallmark lesions, amyloid-beta (Aβ) and neurofibrillary tangles (NFTs) found in the brain at post-mortem [1]. The disease is believed to initiate in the cerebral cortex and medial temporal lobes, with the neocortex (hippocampus) taking the major impact of the neuropathology [2,3]. Research supports the view that NFTs appear first in the subcortical nuclei, such as the Locus Coeruleus (LC) in the pons [4], implying NFT pathology originates in the LC and spreads to the cerebral cortices and the hippocampus. The Braak neuropathological staging of the disease places the accumulation of phosphorylated tau (p-tau) NFTs in the LC and other subcortical nuclei to precede stages I–IV. This occurs both before cognitive impairment and the onset of LC cell loss, which is not significant until Braak stage III [4].

So, why was the LC overlooked in the initial map of AD neuropathology? Perhaps the involvement of the LC historically was considered non-specific to AD as it plays such diverse psychological and physiological roles in the human body, for example, in rapid eye movement sleep disorder, schizophrenia, anxiety, and depression [5,6]. With that said, the rostral and middle loci coerulei regulate sleep-wake cycles [7], attention [8], memory [9], mood and behavior [10]; which correlates with AD clinical behavior to some extent. Alternatively, the caudal LC regulates cardiovascular, respiratory, and gastrointestinal activities [11], which have only recently been associated with AD by modern day research enabling the wider understanding of AD development via comorbidities and microbial dysbiosis [12,13,14,15,16]. The advent of magnetic resonance imaging (MRI) has enabled in-situ visualization of the brainstem nuclei, such as the LC [17], confirming neurodegeneration during the aging process, and during the clinical AD phase, as was previously thought [18,19,20]. Could it be that neurodegeneration involves loss of noradrenergic cells as represented by the deficit in noradrenaline synthesis observed in AD brains compared with age-matched non-AD control brains [21]? Low levels of noradrenaline in the LC negatively affects tight-junction proteins, which, in turn, affect the permeability of the blood-brain barrier (BBB) [22].

To support the concept of the AD NFT lesion being initiated in the brainstem, it can be argued that anatomically, the LC does project to the trans-entorhinal and entorhinal areas of the brain. Therefore, the potential for the pathological tau protein to spread from the LC to the temporal brain remains even if it takes decades before the symptoms of this mental disease appear. Focusing on the LC as being the major norepinephrine producing nucleus, amongst its many roles, noradrenaline is implicated in regulating neuroinflammation as well as reducing oxidative stress [23,24]. Its deficit accounts for an increase in microgliosis, supporting the theory that AD is an inflammatory disease [25,26,27].

The neuropathology of Aβ lesion formation in the LC at the time of NFT tau pathology Braak stage III remains unclear. However, animal, and human studies have shown Aβ pathology to be one mode of LC neuronal loss [28,29]. Topographically the areas of neuronal loss correspond to the localization and distribution of Aβ plaques along the rostral to caudal portions of LC, and to the frontal, temporal, and occipital cortices of the brain [30]. The increased Aβ burden may be attributed to the lack of noradrenaline, or conversely to transmitter loss due to aberrant neurodegeneration, as existing evidence supports both options. However, the mechanisms behind the triggers currently remain unclear. Experiments on N-2-Chloroethyl-N-ethyl-2-bromobenzylamine hydrochloride deprived mice, a feature widely accepted to deplete noradrenaline [31], has demonstrated a five-fold increase in Aβ deposition with an increased size of senile plaques [32]. Surprisingly, other experiments conducted on mice with impaired ability to produce noradrenaline have failed to show increased senile plaque deposition. This suggests that amyloid deposition is unlikely to be from decreased noradrenaline levels; and that potentially cell loss is caused by neurodegenerative processes [33].

## 2. Locus Coeruleus and the Trigeminal Mesencephalic Nucleus

In order to explain the LC neuronal degeneration, there is a need to describe its neuroanatomical connections. LC is part of the so called “Isodendritic Core” or Reticular Formation that, together with the Dorsal Raphe Nucleus, Substantia Nigra and the Meynert Nucleus Basalis in the forebrain, plays an essential role in neuromodulation, regulating behavior and homeostasis by multiple aminergic and cholinergic projections to the cortex [34,35]. Stereological methods have shown the LC to be composed of about 98,000 neurons over a volume of 13 mm^3^ [36], describing mainly two classes of medium sized neurons: small spindle cells and large multipolar cells, the former being densely packed in the rostral portion of the nucleus [37] (Figure 1). The neural connections between the LC and the cerebral cortex are mediated by three different pathways: a monosynaptic direct connection, indirectly through the thalamus, or through the basal forebrain nuclei of the Meynert Nucleus Basalis.

The LC and the subnucleus coeruleus are responsible for the provision of noradrenaline to the brain; in this regard their aminergic connections are projecting to the forebrain, including the hippocampus and the septum, as well as the basal ganglia, cerebellum, and the spinal cord [38] (Figure 1). Alteration of the LC structure causes noradrenaline levels to drop progressively, promoting a neurotoxic proinflammatory milieu, reducing the Aβ clearance and impacting on cognitive function [39].

The trigeminal mesencephalic nucleus (Vmes) is located at the ventro-lateral end of the periaqueductal grey matter and at the lateral end of the floor of the fourth ventricle, just lateral to the LC. The Vmes internal synapses are reactive to neurotransmitters such as noradrenaline, gamma-aminobutyric acid, dopamine, serotonin, and glutamate [40]. This may explain the reason behind the variety of connections that Vmes has with different areas of the brain, including the LC (Figure 1).

Vmes is formed of pseudo-unipolar neurons whose fibers project from masticatory muscles and periodontal proprioceptors [41,42], making it a crucial control hub for chewing processes [43,44,45] and for periodontal proprioception [46]. Evidence shows that 20% of the neurons within the Vmes in cats, [47] and approximately 10–15% in monkeys, display afferent fibers from the mechanoreceptors located in the periodontal ligament (PDL) [48].

The functions related to the dentition, including mastication and pronunciation, are crucial for survival. The observed neurodegeneration caused by tooth loss is not well understood. It would appear to cause damage to the sensory receptors of the pulp and the PDL as encountered in several experimental observations [49,50]. Furthermore, death of neurons within the Vmes appears to occur after peripheral axotomy in rats [51], as well as following tooth extraction in cats [52]. Therefore, intimate spatial and functional interaction between LC and Vmes has led us to consider a potentially fatal involvement of neurons both following the degeneration caused peripherally by a pulpal and periodontal damage, or by infection.

## 3. The Mesencephalic Nucleus (Vmes) and Its Connections with the Oral Structures

The Vmes represents an original and unique example of a cluster of cell bodies of primary afferent neurons from intra-oral proprioceptors and spindles from muscles involved with jaw closure.

Each axon from the Vmes is divided into united, central, and peripheral branches, and each branch projects axon collaterals. The united axon is a fiber extending from the cell body to the T or Y shaped bifurcation point. The central branch is a fiber that extends from the bifurcation point to a caudal direction, always within the brainstem, while the peripheral axon travels from the bifurcation to outside the brainstem. The main collateral branches, which emerge from the united, the central and the peripheral axons can bifurcate further into secondary and tertiary collaterals.

There are two types of Vmes neurons, which are distinguished based on the morphology of the axon and its central projections. The first type of Vmes neuron is pseudo-unipolar and is different from dorsal root ganglion neurons as gives off collaterals from all three axonal segments: united, peripheral, and central. This profile, with a massive collateral system, is reportedly very similar to the Vmes primary afferents from masseter spindles as its central fibres project to supra (Vsup), inter-trigeminal (Vint) and juxta trigeminal (Vjuxta) regions. The main difference is that that they also project to the trigeminal motor nucleus, looping the reflex arch.

The second type of Vmes neuron is bipolar in shape with a peripheral component leading to the periodontal proprioceptors and is double the size of the first type of Vmes neurons [45].

Nakamura et al. [53] examined the ultra-structure of the nerve bundles in the human PDL and observed that some of them were ensheathed with perineurium, while others were not. The nerve bundles near the apical region of the ligaments demonstrate special features; as swollen oval-shaped unmyelinated nerve fibers, distant from blood vessels and distributed in the central area of the PDL, slightly closer to the alveolar bone. Two different sized fibers were visible from these bundles: small fibers (approx. 10 µm) and large fibers (approx. 25 µm). The fibers demonstrated a different organelle content: the smaller fibers were surrounded by a thin Schwann cell contained within a disordered array of microtubules, neurofilaments and scattered mitochondria, the larger fibers possessed the same elements but were richer in mitochondria and dense vacuoles [54]. Fibre density within the PDL varied: the smaller fibers were separated by 4 µm spaces from each other, while the larger were packed into a larger space of 140 µm, within a fibroblast-like capsule. The role of these fibers as proprioceptors within the PDL has been investigated further by Cash and Linden [55]. It would appear that the slowly adapting receptors, mainly the sheathed ones, were firmly attached to the surrounding tissue located in the deeper areas of the ligament, predominantly around the root apices, while the rapidly adapting receptors without the sheath (unmyelinated), were loosely attached [56], located in the middle portion of the periodontal space, and were responsible for the rapid stimuli to the brain stem.

## 4. Tooth Loss Can Cause Central Neuroinflammation and Neurodegeneration

Many clinical observational studies have demonstrated that poor oral hygiene and development of periodontal disease leading to increased tooth loss could be the major risk factors for cognitive impairment and subsequent dementia development and progression [57,58]. The persistence of chronic inflammation [59,60] caused by periodontal disease and the spread of oral bacteria and/or their toxins to the central nervous system [61] have the potential to initiate neurodegeneration in the hippocampus [62], and impair the reflexes involved in the masticatory system [63,64].

The main question is whether prior existence of neurodegeneration encourages a reduced coping strategy of patients to prioritize time to their personal oral care and thus increases the risk of tooth loss. Or alternatively, could it be that oral inflammation potentially acts as a trigger for the delayed neurodegeneration? Further research is needed to clarify these questions.

In vivo animal studies have investigated the neurodegeneration within the Vmes periodontal afferents after maxillary molar extractions [65]. The results demonstrate degeneration of Vmes neurons occurring within 5 days post tooth extraction, followed by death of afferents without any attempts of recovery [50,65]. However, neuronal apoptosis extending from the Vmes to the trigeminal motor nuclei connections was reported. A systematic review based on 26 different studies has shown how the reduction of dental pulp [66] and periodontal disease derived neural signals [67], as is seen in tooth loss, could have a negative impact on cognition related brain regions [68]. The mastication process involves multiple brain areas including the Vmes, the prefrontal cortex, thalamus, the limbic system, and the LC, as well as the communication among different systems (Figure 1) related to cognition [69]. Thus far, investigations involving this mechanical pathway have shown a probable dual relationship. Firstly, the loss of occlusal support can decrease the overall blood supply in the brain and the effects are visible in cognition related areas 12 weeks post tooth extractions in animals [68]. Secondly, the peripheral damage due to molar tooth removal has the potential to reduce the number of new neuron cells in the hippocampus of animals, and this also contributes to cognitive decline [70]. A plausible explanation could be that the weakening of the afferent stimulation with a consequent down regulation of the synaptic plasticity in the hippocampal region and its signaling system is taking place [70]. Even the impaired mastication and muscle activity would appear to be responsible for the neurodegenerative process via poorly understood pathways. Muscle contraction activation leads to the release of Neprilysin, an enzyme that is involved in the clearance of Aβ peptide. An impaired ability to chew due to muscular inability could reduce the axonal transportation to Vmes of this protective factor, allowing for more Aβ to deposit in the pathology prone anatomical areas of the brain and trigger general neuroinflammation [71].

## 5. Locus Coeruleus Degeneration and the Blood-Brain Barrier Impairment

The BBB protects the brain parenchyma with a highly selective interface facilitating the transport of nutrients into the brain and preventing access to pathogenic blood-borne agents including bacteria [72,73]. The structural and functional features include the brain’s endothelial barrier, neurons, perivascular astrocytes, pericytes, microglia and the extracellular matrix. They are all components of the so-called neurovascular unit [74]. The main element in maintaining the barrier function is the tight junctions, formed by primary proteins (claudin, occludin, junctional adhesion molecules and accessory proteins of the zona occludens [75] and the adherens junctions as ZO-1, ZO-2, ZO-3), connecting adjacent endothelial cells via proteins of the cadherin superfamily [76]. BBB disruption, or its increased permeability, has been observed in AD [77], and this could be a plausible mode of neurodegeneration. Dynamic Contrast Enhanced MRI performed in early-stage AD patients compared to age matched negative controls has shown a clear leakage of contrast agents (gadobutrol), which are used in Diagnostic Test for BBB integrity. These contrast agents were in areas of cerebral microbleeds confirming a leaky BBB [78]. Whether the BBB impairment causes neurodegeneration, or the neurodegenerative damage causes the disruption of the BBB is unclear.

The impairment of BBB function could potentially be due to the LC degeneration and can be explained by several non-incompatible scenarios. Firstly, it could be a direct consequence of the alteration in the noradrenergic neurotransmitter transmission implicating the role of noradrenaline in the BBB regulation via cyclic AMP [79] and the tight junction control [80]. An alternative explanation could be that the BBB impairment is due to the progressive alteration of the neuronal system under the control of noradrenaline, which could, therefore, delay the dopaminergic transmission [81,82,83]. This hypothesis could also explain the lack of integrity of the BBB, which is a common feature seen in different neurodegenerative diseases including AD. Finally, the impact on the BBB permeability could be related to the generalized neuroinflammation caused by the impact of dying neurons on the microglial component of the neurovascular unit. Microgliosis and increased levels of inflammatory mediators are present in AD and in the corpus striatum of patients affected with Parkinson’s disease [84,85]. In support of the above observations, research shows that the dysfunction of the noradrenergic fibres and reduction due to LC impairment and degeneration, loss of tight junction proteins in the vascular cell to cell contacts, a 50% decrease in total ZO1 protein, and a relative increase in the beta occludin isoform, leads to diffuse gliosis. Therefore, LC degeneration reported in AD may contribute to this disorder via the BBB dysfunction and/or the altered noradrenaline system [22].

## 6. *Treponema denticola* as a Potential Cue for the Locus Coeruleus Impairment

*Treponema denticola* and *Poprhyromonas gingivalis* are part of the oral microbiome that harbours approximately 700 different bacterial taxa [86]. They belong to the so-called red complex consortium of bacteria, which is commonly associated with the development of the most severe and rapidly progressive periodontitis [87]. They have a symbiotic relationship, whereby their close association can minimise the dilution of metabolites and signalling molecules, facilitating an efficient metabolic communication [88]. In this case, *T. denticola* and *P. gingivalis* display cooperation in protein degradation, nutrient utilisation, growth promotion and motility activation [88,89,90,91,92,93,94]. Evidence from experimental in vivo animal models suggests bacterial invasion from the gingival sulcus/junctional epithelium is a critical process in the pathogenesis of periodontitis [95,96]. Upon their cellular invasion of the epithelium, bacteria are normally segregated within membrane bound vesicles into lysosomes, which degrade/kill them by the action of lysosomal enzymes [97]. Periodontal pathogens such as *P. gingivalis* and *Aggregatibacter actinomycetemcomitans* can survive within the epithelial cells and undergo multiplication to spread to the adjacent cells [98]. *T. denticola* too can invade gingival epithelial cells directly and through the intercellular spaces. Upon internalisation, *T. denticola* can resist endo-lysosomal enzyme mediated degradation and can survive for over 48 h within the toxic milieu that also harbours intracellular reactive oxygen species related stress [97].

Recent evidence demonstrates that *T. denticola* can avoid the host triggering toll-like Receptor-9 (TLR-9) in the endo-lysosomal compartment [99] and inhibits TLR2 signaling axis together with expression of the chemokine interleukin-8 (IL-8) [100] and the human beta defensin [101]. Based on these observations, it is plausible to suggest that *T. denticola* has the potential to reach the peripheral nerve endings in the PDL during stage 3 and 4 of periodontitis post infection spreading and replicating within the infected epithelial cells alongside the active pockets.

Rapid adaptive receptors, such as the smaller fibres, could potentially incite the intra-axonal invasion of *T. denticola* and drive the pathogen to the Vmes and from there to the LC via the above-described neural connections. A similar route has been described for viruses, as the HSV-1 infection, from in vivo and in vitro observations, has been shown to be able to reach the trigeminal ganglion, and from there to reach the trigeminal nuclei in the brainstem, thalamus, and hippocampus of the hindbrain (Figure 1) [102]. It is reported that viral particles could travel anterogradely within the axon, and at fast transport rates (0.8 µm/s). The reason for this could be that the membranes associated with viral particles may contain the receptor that mediates the interactions with axonal anterograde transport machinery. Biochemical analysis of this viral-associated membrane revealed the presence of a cellular amyloid precursor protein, which also co-localized with purified viral particles prior to the in vitro experimental injection into the axon [103].

Another potential theory is that *T. denticola* virulence factors may induce neurotoxic effects or cause impairment of the transport process. Several virulence factors have been described for *T. denticola*, including metabolic end products [104,105,106], protease dentilisin [107], major sheath proteins [108,109], lipoprotein [110] and outer membrane vesicles [111]. The cytotoxic effect of these virulence factors on epithelial cells is fully demonstrated elsewhere supporting this observation [112]. The potential action of *T. denticola* virulence factors may lead to the inhibition of axonal transport by altered localization or delivery of essential cargo, for example, failure to deliver mitochondria to areas of need leading to cell death through energy deprivation. Another effect could be the disruption of lysosomal, autophagosome motility leading to the toxic build-up of aggregated proteins or defective organelles or via an alteration in the cell signaling system [113]. In this regard studies on neuronal cells cultures incubated with *T. Pallidum* have shown an electrophysiological dysfunction after 12 h (lack of action potential response) and a complete cell degradation after 16 h with flattened somata, marked holes on the cytoplasmic membranes, the nuclear profile deteriorated to a coagulated matrix and the cytoskeletal core reduced to small-bleb particles at the edge of the cell profile [114].

The third hypothesis for *T. denticola* to progress to the CNS could be via extra-axonal progression along the nerve sheath. This justifies the potential findings by Riviere in post-mortem autopsies of trigeminal ganglia which showed a build-up of *T. denticola* colonies in dense bundles mimicking pathologic amyloid plaques in AD patients [115]. Electron micrograph evidence from genital syphilomas biopsies on patients, was able to clearly show T. Pallidum colonies in epi, peri and endo-nervium. The spirochaetes were located in the space between the Schwann cells and their basal lamina of both myelinated and unmyelinated axons, without anyone engulfed into the Schwann cell’s cytoplasm, nor within the axons [116].

Recent studies involved mice that were orally infected with *P. gingivalis* and *T. denticola* to clarify the role of the latter bacterium in the potential pathogenesis of AD. The study demonstrated that *P. gingivalis* and *T. denticola* could enter the brain predominantly through the blood circulation and cause Aβ aggregation, however *T. denticola* 16s rDNA was also detected in a small number of trigeminal nerve ganglia, suggesting this pathogen may also directly enter the hippocampus via both blood circulation and the trigeminal nerve where it has the potential to produce Aβ1-40 and Aβ1-42 [117]. Next generation sequencing studies confirm *T. denticola* detection in human AD brains [118,119], the question arises as to whether their pathogenic synergy seen in sub-gingival biofilms is retained in extra oral organs such as the brain involving neurodegeneration of the LS as hypothesized by Pisani et al. [120].

## 7. Discussion

The LC is known to be involved in several psychological and physiological roles with its rostral and middle parts regulating the sleep-wake cycles [121], attention [122], memory [123], mood and behavior [124]; and the caudal part regulating cardiovascular, respiratory, and gastrointestinal activities [125]. Recent evidence shows its main involvement in neurodegenerative diseases, such as AD, to be in the pre-symptomatic phases, due to preceding tau pathology and extensive neuronal loss. As discussed earlier, this weakness in the LC may be due to circulating noxious substances or due to the lack of myelinization of its neurons. Therefore, this renders the brain more susceptible to oxidative stress and the effects of neuroinflammation. The proximity of the LC with the brainstem trigeminal nuclei, such as the Vmes, has recently been evaluated in terms of a potential interaction between them in the pathogenesis of neuroinflammation.

Recent studies have shown that the nerve endings of Vmes neurons are distributed in the middle and apical part of the PDL, as well as in spindles of the masticatory muscles [44]. In addition, degeneration of Vmes afferents can lead to full neuronal loss in the nucleus after either peripheral axotomy [51] or tooth extraction [50]. Conversely, cohort studies on patients affected by periodontal disease have shown that poor oral and gingival health together with disease severity represents a strong risk factor for progression of cognitive impairment [126,127]. In addition, the rate of tooth loss and cognitive function are closely related [128].

In a clinical study conducted on saliva samples from 20 patients diagnosed with AD, almost 35% showed a significant presence of periodontal pathogens including *P. gingivalis* and *T. denticola*. While a significant and a clear correlation has been shown between salivary *P. gingivalis* concentration and lower cognition; *T. denticola* appeared to decrease the serum concentration of neopterin and indirectly the concentration of kynurenine and tryptophan, an amino acid involved in synaptic neurotransmission [129].

Only one German trial has investigated the effects of periodontal treatment on a preclinical AD population. The aim was to quantify the average treatment effect among the treated or the within subject difference by comparing outcomes between treated against the estimated potential outcome of untreated individuals [130]. Periodontal treatment was shown to be effective in stopping the progression of AD related brain atrophy (−0.41; 95% C.I.: −0.70; −0.12) with a robust significance, but it remains unclear whether the treatment had been effective via the prevention of periodontal bacteremia inciting spread to the brain, or if it had simply reduced the local and systemic inflammatory parameters leading to generalized neurodegeneration.

The aim of the present narrative review was to collect and analyze the evidence regarding the connection between periodontal infection and/or damage and the LC neurodegeneration. A robust experimental and clinical library of evidence correlates LC atrophy, AD and indirectly the neuronal degeneration of Vmes fibres from the PDL. This degeneration observed may be due to tooth removal, due to experimental axotomies, or due to the action of peripheral infectious agents such as *P. gingivalis* and *T. denticola*. These bacteria, before entering the blood stream via bacteremia, may interact peripherally with the periodontal nerve endings and may either enter the axon with an endocytic action, then progressing along the fiber up to the Vmes, or by causing axonal damage disrupting the cargo system or mitochondrial deprivation with subsequent cell starvation and death. The defective BBB permeability and the effects of the circulating *P. gingivalis* LPS also appear to play a role in AD progression: the early involvement of the LC would likely cause the norepinephrine levels to decrease and would negatively affect the tight junctions and the overall neurovascular unit.

## 8. Conclusions

Research-based evidence provides vital information as to how periodontal disease and neurodegenerative diseases may be comorbid. The shared neural pathways, the neurotropism of the main periodontal pathogens such as *P. gingivalis* and *T. denticola,* and the remarkable neuroinflammatory effects of the ongoing periodontal lesion and the consequences of tooth loss have highlighted the importance of screening for disease and providing treatment and appropriate follow up for these patients.

The LC has been considered to be a critical station for the neurodegenerative process as it displays evidence of neuronal loss and active microgliosis in the earliest stages of AD, and the observed norepinephrine deficit would appear to be the most reasonable cause for the reduction of cortex functionality and the increase in BBB permeability.

The Trigeminal Vmes, displaying the effects of neuronal loss due to the peripheral impairment of their axons in the PDL, could potentially spread damage to the adjacent LC via the intimate neural pathways that connect them and from there up to the hippocampus and then on to affect the entire brain.

While the role of *P. gingivalis* and its toxins in AD is widely reported, there is a lack of research about *T. denticola*. There is, therefore, a need to explore the role of this spirochete in the processes involved in peripheral nerve damage which subsequently leads to neuroinflammation following periodontal infection. Further research is required within this field from a clinical point of view to explore how periodontal treatment may be effective in preventing neurodegenerative diseases, or whether it could somehow slow down the neurodegenerative process thus delaying the onset of debilitating diseases such as AD.

There is some hope on the horizon in terms of a potential therapeutic route to reduce AD symptoms based on noradrenergic pathways. Historically, AD therapy has targeted boosting the acetylcholine system, but the benefits remain limited. Recently, research has explored drugs with noradrenergic action to treat the cognitive symptoms and apathy associated with AD. These drugs currently are used to treat other conditions such as attention deficit-hyperactivity disorder or depression, but a systematic review and meta-analysis [132] shows promising signs of these agents providing relief of some of the debilitating symptoms experienced in AD via inhibiting the reuptake of noradrenaline, thus prolonging its effect. This would appear to improve memory and apathy. For so long, a diagnosis of AD incited a hopeless prognosis and inevitable outcome, but finally thanks to research improving our understanding of disease mechanisms, new therapeutic pathways are being explored which provides us with some hope in the relentless battle against this relentless mental disease. In addition, maintaining an oral microbiome symbiosis and preventing any periodontal disease with regular surveillance and good oral hygiene throughout life is likely to reduce the incidence of AD. The pathways to neurodegeneration and/or neural damage via periodontal disease presented here are significant adding to our growing knowledge of causal associations between dysbiotic sub-gingival biofilm under the influence of *T. denticola* and its highly synergistic keystone pathogen *P. gingivalis* and the early development of AD.

## Figures and Tables

**Figure 1 ijerph-20-01007-f001:**
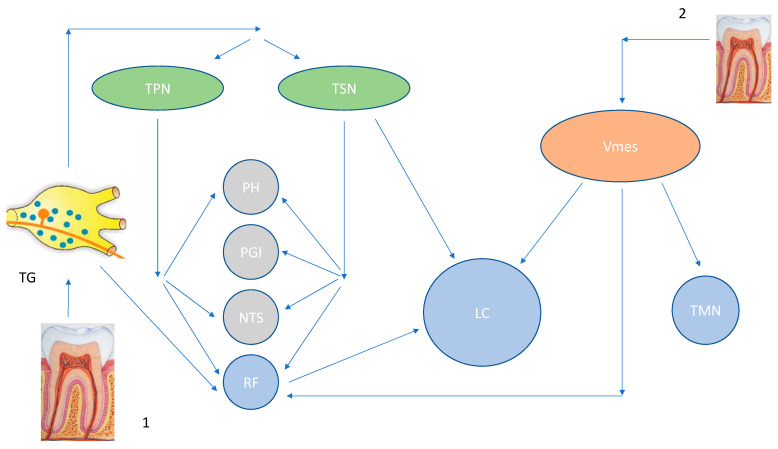
[131]: Trigeminal Pathways to Brainstem structures. Legenda: TG: Trigeminal Ganglion, TPN: Trigeminal Principal Nucleus, TSN: Trigeminal Spinal Nucleus, PH: Prepositus Hypoglossi, PGI: Nucleus Paragigantocellularis, NTS: Nuclues of Tractus Solitarius, RF: Reticular Formation, Vmes: Trigeminal Mesencephalic Nucleus, LC: Locus Coeruleus, TMN: Tuberomammillary Nucleus, 1: Somato-sensory and Periodontal Ligament/Pulpar afferents, 2: Proprioceptory Periodontal Ligament afferents.

## Data Availability

Not applicable.

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
