# Peer review of "Locus Coeruleus Dysfunction and Trigeminal Mesencephalic Nucleus Degeneration: A Cue for Periodontal Infection Mediated Damage in Alzheimer’s Disease?"

_ijerph, 2023, doi:10.3390/ijerph20021007_

Round 1

Reviewer 1 Report

General considerations

The review by Pisani et al. links two interesting fields of research, the former attributing the onset of neurodegenerative diseases to degenerative process occurring at the level of noradrenergic Locus Coeruleus (LC) neurons, and the second which correlate brain degenerative processes to tooth loss and poor oral care. Within this frame the authors unify the two topics proposing that the pathological LC changes, which trigger neurodegenerative diseases may  arise from a bacterial infection which starts from the mouth and reach the LC trough the peripheral and the central trigeminal pathways. The argument of the review is of extreme interest. I would suggest the Authors to take into accounts papers documenting transport of bacteria and or of their toxins along nerve fibres (see specific comments). Moreover, they have to consider that the link between teeth loss and LC degeneration may be attributable to loss of the trophic factors reaching the LC from trigeminal peripheral tissues and afferents, which are known to control LC neurons development/ differentiation and can exert an import role also in adult age. Finally, trigeminal afferents seem also to acutely affect LC discharge and cognitive performance. Chewing enhances cognitive performance, while a trigeminal asymmetry, which is associated to an asymmetric pupil size, suggesting the presence of an asymmetry in LC discharge, is detrimental to it. A prolonged imbalance in LC activity might raise too much the basal LC discharge on one side, a phenomenon which has been proposed as facilitator of degenerative processes (Long Road to Ruin: Noradrenergic Dysfunction in Neurodegenerative Disease. Weinshenker D. Trends Neurosci. 2018 Apr;41(4):211-223. doi: 10.1016/j.tins.2018.01.010). Beyond these points that could be addressed in the review, the text is often unclear and the quoted paper are not adequately explained. For these reasons, a thoroughly revision is necessary before accepting the paper for publications.

Specific comments

1. Introduction

 Page 2, line 14

“Alternatively, the caudal LC regulates cardiovascular, respiratory, and gastrointestinal activities (11),”.

The caudal LC is also associated to postural control: norepinephrine enhances motoneuron excitability and the drop of LC discharge is associated to postural atonia: is there any evidence of reduced postural tone in Alzheimer disease?

Page 2, line 38

Could you clarify the term “aberrant neurodegeneration”? Is there an ordinary neurodegeneration?

Page 2, line 42

“Surprisingly, other experiments conducted on mice with impaired ability to produce noradrenaline have failed to show in-creased senile plaque deposition. This suggests that amyloid deposition is unlikely to be from decreased noradrenaline levels; and that potentially cell loss is caused by neuro-degenerative processes (33)” .

As proposed by Mara Mather, the relation between Noradrenaline and beta amyloid plaques can be more complicated (Noradrenaline in the aging brain: Promoting cognitive reserve or accelerating Alzheimer's disease? Mather M. Semin Cell Dev Biol. 2021 Aug;116:108-124. doi: 10.1016/j.semcdb.2021.05.013.)

2. Locus Coeruleus and the Trigeminal Mesencephalic Nucleus

I suggest to include in this section that the connections of trigeminal afferents with the LC explains interesting phenomena which could be related to trigeminal induced degeneration of LC neurons. Sensorimotor trigeminal activity acutely improves cognitive performance (Short-Term Effects of Chewing on Task Performance and Task-Induced Mydriasis: Trigeminal Influence on the Arousal Systems.Tramonti Fantozzi MP, De Cicco V, Barresi M, Cataldo E, Faraguna U, Bruschini L, Manzoni D. Front Neuroanat. 2017 Aug 8;11:68. doi: 10.3389/fnana.2017.00068). The Author may also consider that cognitive performance increases (The path from trigeminal asymmetry to cognitive impairment: a behavioral and molecular study. Tramonti Fantozzi MP, Lazzarini G, De Cicco V, Briganti A, Argento S, De Cicco D, Barresi M, Cataldo E, Bruschini L, d'Ascanio P, Pirone A, Lenzi C, Vannozzi I, Miragliotta V, Faraguna U, Manzoni D. Sci Rep. 2021 Feb 26;11(1):4744. doi: 10.1038/s41598-021-82265-6) and skilled movements are more easy to control (Unbalanced Occlusion Modifies the Pattern of Brain Activity During Execution of a Finger to Thumb Motor Task.Tramonti Fantozzi MP, Diciotti S, Tessa C, Castagna B, Chiesa D, Barresi M, Ravenna G, Faraguna U, Vignali C, De Cicco V, Manzoni D. Front Neurosci. 2019 May 17;13:499. doi: 10.3389/fnins.2019.00499. eCollection 2019) following occlusal rebalancing, which makes the LC activity more symmetric. An asymmetric LC discharge may rise too much the activity on one side and this may represent the trigger for neurodegeneration (Long Road to Ruin: Noradrenergic Dysfunction in Neurodegenerative Disease. Weinshenker D. Trends Neurosci. 2018 Apr;41(4):211-223. doi: 10.1016/j.tins.2018.01.010. Moreover, in the early development trigeminal fibres supply LC neurons with important factors fundamental for their differentiation (Onecut factors control development of the Locus Coeruleus and of the mesencephalic trigeminal nucleus (Espana A, Clotman F. Mol Cell Neurosci. 2012 May;50(1):93-102. doi: 10.1016/j.mcn.2012.04.002). The trigeminal system supply LC neurons with trophic factors also in juvenile and adult age (quoted in ref 132) and lack of this trophic support may cause neurodegeneration in addition to the microbial infection proposed by the Authors.

Page 3, line 2

The neural connections between the LC and the cerebral cortex are mediated by three different pathways: a monosynaptic direct connection, indirectly through the thalamus, or through the basal forebrain nuclei of the Meynert Nucleus Basalis.

A reference is needed.

Page 3, lines 14-17

The Vmes internal synapses are reactive to neurotransmitters such as noradrenaline, gamma-aminobutyric acid, dopamine, serotonin, and glutamate (40). This may explain the reason behind the variety of connections that Vmes has with different areas of the brain, including the LC (Figure 1).

What the Authors mean for “internal synapses”? Afferent synapses on Vmes neurons? And why the presence of neurotrasmitters, likely released by afferent pathways to Vmes neurons, should explain the variety of efferent connections?

Page 3, lines  25- page 4, line 6

The observed neurodegeneration caused by tooth loss is not well understood……..”

 Are the authors talking about Vmes neurons degeneration following teeth extraction? Please clarify the issue and improve the text. If peripheral nerve lesions leads to degeneration of Vmes neurons and this finding suggests the possibility of a trans neural degeneration of LC neurons this have to be stated more clearly.

3. The Mesencephalic Nucleus (Vmes) and its connections with the oral structures

I wonder whether it is important to devote a section to histological details on the Vmes neurons morphology and PDL structure. They are not useful for clarifying the topic argument of this review

Page 4, line 24

The main difference is that that they also project to the trigeminal motor nucleus, looping the reflex arch.

Is not clear what the authors are comparing. The first type of Vmes neuron with the cell body of identified spindle afferents? If the central projections of the two neurons do not correspond, what is the role of the first type of Vmes neurons? Please clarify.

4. Tooth loss can cause central neuroinflammation and degeneration

Page 5, line 3

and impair the reflexes involved in the masticatory system (63; 64)”.

What are the reflexes the Authors are talking about? The quoted papers (63,64) seem to be addressing the relation between poor mastication and cognitive deficits, not that of masticatory reflexes. Please clarify.

Page 5, lines 5-9

The main question is whether prior existence of neurodegeneration encourages a reduced coping strategy of patients to prioritise time to their personal oral care and thus increases the risk of tooth loss. Or alternatively, could it be that oral inflammation potentially acts as a trigger for the delayed neurodegeneration? Further research is needed to clarify these questions.

This is an important question. In my opinion it should be emphasised that animal experiment indicates that:

1) teeth extraction or teeth grinding leads to neurodegeneration and cognitive deficits and,

2) in teeth grinding experiments both cognitive deficits and neurodegeneration are partially prevented by restoring the occlusal surface (Effect of occlusal rehabilitation on spatial memory and hippocampal neurons after long-term loss of molars in rats. Sakamoto S, Hara T, Kurozumi A, Oka M, Kuroda-Ishimine C, Araki D, Iida S, Minagi S. J Oral Rehabil. 2014 Oct;41(10):715-22. doi: 10.1111/joor.12198).This evidence favours the hypothesis that oral troubles (inflammation or what else) trigger neurodegeneration.

Page 5, lines 29-30

“Even the impaired mastication and muscle activity would appear to be responsible for the neurodegenerative process via poorly understood pathways.”

It seems to me that the connection of Vmes with the LC may well explain this neurodegeneration

5. Locus Coeruleus degeneration and the Blood-Brain Barrier Impairment

Page 6, lines 3-6

“An alternative explanation could be that the BBB impairment is due to the progressive alteration of the neuronal system under the control of noradrenaline, which could, therefore, delay the dopaminergic transmission (81; 82; 83).”

The relation of dopamine to BBB has not been addressed. The reader should not be obliged to read ref 81-83 for understanding your message. Please clarify.

Page 6, lines 11-15

“In support of the above observations, research shows that the dysfunction of the noradrenergic fibres and reduction due to LC impairment and degeneration, loss of tight junction proteins in the vascular cell to cell contacts, a 50% decrease in total ZO1 protein, and a relative increase in the beta occludin isoform, leads to diffuse gliosis.”

Please improve this sentence. “Reduction” of what? I think leads should be lead.

Page 6, lines 16-17

Therefore, LC degeneration reported in AD may contribute to this disorder via the BBB dysfunction and/or the altered noradrenaline system (86).

I suggest:  may contribute to AD symptoms as a consequence either of the BBB dysfunctions or else of the disrupted release of noradrenaline on brain structures.

6. Treponema denticola as a potential cue for the locus coeruleus impairment

Pag 6 lines 44-46

Rapid adaptive receptors, such as the smaller fibres, could potentially incite the intra-axonal invasion of T. denticola and drive the pathogen to the Vmes and from there to the LC via the above-described neural connections.

It worth of note that both bacteria (Axonal transport of Listeria monocytogenes and nerve-cell-induced bacterial killing. Dons L, Jin Y, Kristensson K, Rottenberg ME. J Neurosci Res. 2007 Sep;85(12):2529-37. doi: 10.1002/jnr.21256.) and their toxins (Uptake of Clostridial Neurotoxins into Cells and Dissemination. Connan C, Popoff MR. Curr Top Microbiol Immunol. 2017;406:39-78. doi: 10.1007/82_2017_50) can be retrogradely transported in the axons.

7. Discussion

Page 8, lines 25-29

These bacteria, before entering the blood stream via bacteremias, may interact peripherally with the per-iodontal nerve endings and may either enter the axon with an endocytic action, then pro-gressing along the fibre up to the Vmes, or by causing axonal damage disrupting the cargo system or mitochondrial deprivation with subsequent cell starvation and death.”

See previous comment

Author Response

Dear Reviewer,

many thanks for your perusal and useful thoughts.

All the amendments have been done changing the MS according to your reflections including the Title.

However, regarding the suggested papers, we carefully read, these are our comments about their fit in our review topic:

Uptake of Clostridial Neurotoxins into Cells and Dissemination by Chloé Connan and Michel R. Popoff is irrelevant for the message in our ms because Clostridium is not part of the oral biofilm.

Mini-Review Axonal Transport of Listeria monocytogenes and Nerve-Cell-Induced Bacterial Killing by Lone Dons,1 Yuxuan Jin, Krister Kristensson, and Martin E. Rottenberg 2007.
Listeria monocytogenes (L. monocytogenes) mediated neuronal damage. This infection is related to enchepalitis and this is not part of Alzheimer’s disease pathology. therefore irrelevant to our story.  Paper itself is very interesting.

Onecut factors control development of the Locus Coeruleus and of the mesencephalic trigeminal nucleus A. Espana, F. Clotman  2012.
This describes embryonic aspect of locus coeruleus again is irrelevant to our message as we are talking about an aged brain which has suffered from neurodegeneration as in Alzheimer’s disease.

Many thanks again for your help.

Reviewer 2 Report

The review by Pisani and colleagues describes a very novel hypothesis that attempts to link together the growing appreciation of periodontal disease and increased dementia risk with locus coeruleus damage as another indicator of neurodegenerative disease. The content is clearly presented and a quite novel association is described. Although the mechanistic link is still tenuous the literature is not overinterpreted and the review is more than simply a restatement of the field but instead a thought provoking interpretation.

Author Response

Thank you so much for your review. We reviewed the grammar forms at our best.

Best Regards

Reviewer 3 Report

Congratulation for tyour narrative review

Please in the conclusion can you add what do you think could be the role of dentist / periodontist ti reduce AD risk  correlated to oral infectious ? It's very important to add a sentence

Best regards

Author Response

Thanks a lot for the inspiring review. We added the relevance of periodontal monitoring in the Conclusion part.

Reviewer 4 Report

1. The objective of the study is good and novel.

2. Figure 1 can be redesigned explaining the process more clearly.

3. Figures are missing in some sections. It would be more understandable for readers if mechanism is explained through figures.

4. Include tables for invitro and preclinical work.

5. Include clinicals if available.

Author Response

Thanks a lot for your review.

Unfortunately the lack of clinical or experimental data in this regards won't allow to display tables. The figure contains a clear caption where all the different euro-stations and the different connections are mentioned. The overall explanation is within the ms, therefore we thought it was redundant to make our figure too wordy. Many thanks.